# Effect of the COVID-19 Pandemic on Medical Student Career Perceptions: Perspectives from Medical Students in China

**DOI:** 10.3390/ijerph18105071

**Published:** 2021-05-11

**Authors:** Carla Zi Cai, Yulan Lin, Haridah Alias, Zhijian Hu, Li Ping Wong

**Affiliations:** 1Department of Epidemiology and Health Statistics, School of Public Health, Fujian Medical University, Fuzhou 350122, China; carlacaizi@gmail.com; 2Centre for Epidemiology and Evidence-Based Practice, Department of Social and Preventive Medicine, Faculty of Medicine, University of Malaya, Kuala Lumpur 50603, Malaysia; haridahalias@gmail.com

**Keywords:** medical student, career turnover, COVID-19, social support

## Abstract

Our aim was to examine perceived occupational turnover intentions among medical students and the associated factors. A cross-sectional study using a Web-based survey was conducted. A total of 2922 completed responses were received (response rate 55.7%). A total of 58.4% (95% CI 56.6–60.2) reported high turnover intention (score of 7–15). The odds of higher total turnover score among the fifth-year students was nearly four times that of first-year students (OR = 3.88, 95% CI 2.62–5.73). Perception of the medical profession as not being of high social status and reputation significantly influenced high turnover intention scores (OR = 2.26, 95% CI 1.90–2.68). All three dimensions of the multidimensional scale of perceived social support (MSPSS) significantly predict turnover intention. Lower scores in the support from Significant Other (OR = 1.47, 95% CI 1.17–1.84), Family (OR = 1.47, 95% CI 1.18–1.83) and Friend (OR = 1.42, 95% CI 1.14–1.77) subscales were associated with higher turnover intention. Low score in the Brief Resilience Scale (BRS) was also associated with higher turnover intention (OR = 1.44, 95% CI 1.17–1.77). The findings shed light on the importance of changing public attitudes towards respecting the medical profession and improving the implementation of policies to protect the well-being of people in the medical profession.

## 1. Introduction

The novel coronavirus (COVID-19) infection was first reported in Wuhan, China in December 2019, and three months later the World Health Organization (WHO) declared the spread of the novel coronavirus COVID-19 a pandemic. As of the end of December 2020, there have been over 79.2 million cases and over 1.7 million deaths reported since the start of the pandemic [1]. Healthcare professionals play a critical role in providing intensive care during the pandemic. In countries that are severely impacted by COVID-19, the surge in cases has led to dwindling healthcare resources and an exhausted workforce [2,3]. In China, the shortage of healthcare professionals is an issue that has long been at the forefront of the healthcare industry prior to the COVID-19 pandemic. A high attrition rate among medical graduates and physicians in China has been evident over the past decade [4,5]. The COVID-19 pandemic has exacerbated the shortage of healthcare professionals in China, where the shortages were extremely crucial during the peak of the pandemic [6]. Shortage of healthcare resources may hinder the delivery of high-quality patient care and have adverse effects on patient outcomes and mortality. The critical importance of having a sufficient healthcare workforce during the COVID-19 pandemic warrants investigation on issues surrounding the decline in graduates entering the healthcare workforce. As the need for medical doctors continues to increase with the world facing an unprecedented global health threat, medical students are an important future workforce. Identifying facilitators and barriers influencing turnover intention among medical students will help tailor activities in ways to promote students entering the medical workforce.

Since the pandemic began, high COVID-19 infection rates and deaths among healthcare workers have been reported around the globe [7]. The highly contagious novel coronavirus has resulted in constant fear among healthcare workers of getting the infection and spreading the disease to their families. Fear of infection coupled with intensive work has impacted the psychological and mental well-being of healthcare workers, resulting in decreased job satisfaction, and triggering turnover intention [8,9,10,11]. Findings of a recent systematic review across 15 countries revealed immense mental health problems facing healthcare workers, including depression, anxiety, acute stress reaction, post-traumatic stress disorder, insomnia and occupational burnout throughout the pandemic [12]. In addition to work-related psychological consequences and fear of contagion, in China, the lack of social recognition and social status of medical doctors and low remuneration were found to be additional factors resulting in poor job satisfaction and turnover intention [13,14].

Conversely, several factors were found to have a potentially important protective role in turnover intention. A growing body of research has highlighted social support boosts career satisfaction and adaptability, while mitigating career turnover intentions [15,16]. Resilience has also gained considerable importance in influencing turnover intentions. Being resilient helps manage adverse work situations, overcome hardship and problems at work and reduce negative emotions [17]. A large amount of evidence shows that resilience plays a vital role in enhancing job satisfaction and mitigating the turnover intention among healthcare workers [18,19,20]. A recent study revealed that higher resilience protected healthcare workers from stress during the COVID-19 pandemic [20].

As the need for medical doctors continues to increase with the world facing an unprecedented global health threat, medical students are an important future workforce. Hence, it is important that medical students enter the workforce to fill the serious medical doctor workforce shortage. Therefore, there is a strong need to study medical students’ career perceptions and their interest in clinical careers to project the future growth of the medical doctor workforce. This study aims to examine the perceived occupational turnover intention among medical students and identify factors that influence turnover intention. In light of the preceding discussion, we hypothesized that medical students who perceived higher levels of fear in relation to being in a medical career during the COVID-19 pandemic would be more likely to express higher turnover intention. Additionally, we also hypothesized that attitudes towards the healthcare profession, social support and resilience would promote medical career retention in healthcare facilities, such that positive attitudes, higher levels of social support and resilience would be associated with greater medical career interest. This study may contribute to the understanding of medical students’ career intentions in the era of the COVID-19 pandemic and provide useful insights for the government authorities to prevent further loss of the medical workforce in the healthcare service.

## 2. Materials and Methods

### 2.1. Study Participants and Design

Study participants were all undergraduate medical students enrolled at the Fujian Medical University, Fuzhou, China. The anonymous survey was self-administered online with the link to the questionnaire being sent to all registered medical students. In total there were 5249 medical students in years 1 to 5 at the university.

### 2.2. Measures

The survey consisted of six sections, which assessed (i) students’ demographic background, (ii) fear of COVID-19, (iii) perceived social support, (iv) resilience, (v) attitudes towards the medical profession, and (vi) turnover intention.

#### 2.2.1. Perceived Fear of COVID-19 

Fear of the COVID-19 pandemic was assessed by a self-developed questionnaire. Students were questioned on their perceived level of fear in relation to being in a medical career during the pandemic. A five-item unidimensional scale was answered by preclinical and clinical medical students using a 4-point Likert scale ranging from 0 (not afraid at all) to 3 (very afraid). The composition score ranged from 0 to 15, with a higher scale indicating greater fear of COVID-19. 

#### 2.2.2. Multidimensional Scale of Perceived Social Support (MSPSS)

Perceived social support was measured using the 12-item multidimensional scale of perceived social support (MSPSS) [19]. The MSPSS is comprised of three subscales: Significant Others (items 1, 2, 5, and 10), Family (items 3, 4, 8, and 11), and Friends (items 6, 7, 9, and 12). Every item uses a 7-point Likert scale ranging from 1 (very strongly disagree) to 7 (very strongly agree). An MSPSS subscale score in the range of 5.01–7.00 is considered as high social support [21]. The total possible score ranges from 12–84. A higher score indicates greater social support perceived by an individual. The MSPSS had good reliability (with a Cronbach’s alpha of 0.85 to 0.91) [19]. In this study, the Chinese version of MSPSS was used, and likewise, the scale has been demonstrated to have excellent internal consistency and validity [22,23]. 

#### 2.2.3. Brief Resilience Scale (BRS)

Resilience or the ability to cope with difficulties and recover from the perceived levels of stress during the pandemic among students was measured with a brief resilience scale by Smith et al. 2008 [24], using a six-item scale with a 5-point Likert format ranging from 1 (strongly disagree) to 5 (strongly agree). The summing up gives an overall resilience score of between 6 and 30. Dividing the sum by the total number of questions answered (6) will provide a final BRS score. A score of 1.0–2.99 is considered low resilience, 3.00–4.30 considered normal resilience and 4.31 and above is considered as having high resilience [25,26].

#### 2.2.4. Attitudes towards the Medical Profession 

Attitudes towards the medical profession were assessed with three-item self-developed statements based on previous literature [27,28]. The statements are: ‘Doctor is not a profession of high social status and reputation’, ‘The contributions and sacrifices made by doctors far outweigh their incomes’, and ‘Doctors often experience long working hours with tremendous workload’. Response options were ‘strongly disagree’, ‘disagree’, ‘agree’ and ‘strongly agree’. The item statements were not scored.

#### 2.2.5. Turnover Intention Scale

A modified version of the turnover intention assessment scale by Camman, Fichmen, Jenkins and Klesh (1979) was used to measure the perceived turnover intention [29]. The questions were rephrased as such that the turnover intention is related to medical profession. Turnover intention was assessed by the item ‘I often think of not going into the medical profession in the future’, ‘It is very possible that I will look for a career in the healthcare industry that has zero contact with patients in the future’ and ‘If I could choose again, I would choose not to be on the medical course’. The answer options consist of a 5-point Likert format (extremely disagree, disagree, neutral, agree, extremely agree). The total score ranged from 3 to 15, with a higher score indicating greater turnover intention.

### 2.3. Statistical Analyses

A chi-square test was employed to assess whether there was a significant association between categorical variables. Univariable and multivariable logistic regression analyses were performed to investigate factors influencing turnover intention. All variables found to have a statistically significant association (two-tailed, *p*-value < 0.05) in the univariate analyses were entered into multivariable logistic regression analyses via forced-entry method [30]. Odds ratios (OR), 95% confidence interval (95% CI) and *p*-values were calculated for each independent variable. The model fit was assessed using the Hosmer–Lemeshow goodness-of-fit test [31].

### 2.4. Ethical Considerations

This study was approved by the Medical Ethics Committee at the Fujian Medical University, Fuzhou, China (FJMU 2020 No.1). Students were informed that their participation was voluntary, and consent was implied upon completion of the questionnaire. All responses were collected and analyzed without identifiers.

## 3. Results

A total of 2922 out of a total sample of 5249 registered medical students (response rate 55.7%) completed the survey from 23 November 2020 to 6 December 2020. The proportion of responses was lowest from students in the final year (7.0%). The complete details of students’ characteristics are shown in Table 1. The majority of the participants were Han Chinese (97.1%) and with an average monthly family income of CNY 4000–9999 (49.2%). 

### 3.1. Perceived Fear of COVID-19 

Figure 1 shows the responses to perceived fear of being in a healthcare career during the COVID-19 pandemic. Nearly 63% reported being very afraid/moderately afraid of infecting family members, followed by fear of losing life (34%). The Cronbach’s alpha for the five-item fear of COVID-19 scale was 0.830. The mean and standard deviation (SD) for the total fear of COVID-19 score was 6.1 (SD ± 3.2). The median was 6.0 (inter quartile range (IQR) 5 to 8). The mean fear scores were categorized as a score of 6–15 or 0–5, based on the median split; as such, a total of 1596 (54.6%; 95% CI 52.8 to 56.4) were categorized as having a score of 6–15 and 1326 (45.4%; 95% CI 43.6 to 47.2) were categorized as having a score of 0–5 (Table 1). 

### 3.2. Multidimensional Scale of Perceived Social Support (MSPSS)

Responses to the MSPP are shown in Figure 2. The proportion of agreeing responses for all the items was above 60%. The internal consistencies of the entire scale were good, with a Cronbach’s α of 0.960. The Cronbach’s α of the subscales Friends, Family and Significant Others was 0.942, 0.908 and 0.897, respectively. The mean score (SD) of MSPSS of overall study participants was 5.2 ± 1.1. The proportion of participants reporting scores of 5.01–7.00 was 53.2% (95% CI 51.4–55.0), 57.5% (95% CI 55.6–59.3) and 49.0% (95% CI 40.7–44.4) for subscales Friends, Family and Significant Others, respectively (Table 1).

### 3.3. Brief Resilience Scale (BRS)

The proportion of responses for strongly agree/agree in the BRS is shown in Figure 3. Only two items in the scale reported over 50% of ‘strongly agree/agree’ responses, namely ‘I tend bounce back quickly after hard times’ and ‘It does not take me long to recover from a stressful event’. The Cronbach’s α of the BRS in this study was 0.720. The mean score (SD) of BRS of overall study participants was 3.2 ± 0.6. A total of 20.4% (95% CI 18.9–21.9) reported a BRS score of 1.00–2.99, 76.7% (95% CI 75.1–78.2) reported a BRS score of 3.00–4.30 and only 3.0% reported a BRS score of 4.31–5.00 (95% CI 2.4–3.7) (Table 1).

### 3.4. Attitudes towards the Medical Profession 

As shown in the first column of Table 1, a total of 38.9% ‘strongly agree/agree’ that being a doctor is not a profession with high social status and reputation. The majority ‘strongly agree/agree’ that the contribution and sacrifices made by doctors far outweigh their incomes (84.2%) and that doctors often experience long working hours and tremendous workload (92.5%).

### 3.5. Turnover Intention Scale 

The proportion of responses on turnover intention is shown in Figure 4. A total of 15.9% ‘extremely agree/slightly agree’ that they would choose not to be on a medical course if given a choice, often think of not going into the medical profession in the future (12.5%), and are considering the possibility of venturing into a healthcare industry that has zero contact with patients (10.5%). The mean and SD for the turnover score was 7.2 (SD ± 2.6). The median was 7 (IQR 5 to 9). The mean turnover score was categorized as a score of 7–15 or 3–6, based on the median split; as such, a total of 1706 (58.4%; 95% CI 56.6 to 60.2) were categorized as having a score of 7–15 and 1216 (41.6%; 95% CI 39.8 to 43.4) were categorized as having a score of 3–6. Figure 5 shows the distribution of turnover intention score.

As shown in Table 1, turnover intention was reported highest among the fifth-year students. A total of 79.6% of fifth-year students reported a total score of 7–15 compared to only 43.4% in the first-year students. Year of study is a strong significant predictor of turnover intention. Multivariable logistic regression analysis revealed that turnover intention gradually increased from students from study year 1 to year 5. The odds of a higher total turnover score among the fifth-year students was nearly four times that of the first-year students (OR = 3.88, 95% CI 2.62–5.73). Although a significantly higher proportion of participants from lower household income reported higher turnover intention scores in the univariate, the association is not significant in the multivariable analysis.

Participants who reported a higher fear of COVID-19 score were found to have significantly higher turnover intention scores (OR = 1.39, 95% CI 1.78–1.63). All the three dimensions of the social support in the MSPSS significantly predict turnover intention. Lower scores of the Significant Other (OR = 1.47, 95% CI 1.17–1.84), Family (OR = 1.47, 95% CI 1.18–1.83) and Friend (OR = 1.42, 95% CI 1.14–1.77) subscales were associated with a higher turnover intention score. Similarly, a low resilience score (OR = 1.44, 95% CI 1.17–1.77) was also associated with a higher turnover intention score.

Of the three items of attitudes towards the medical profession, only perceived doctor as not being a profession of high social status and reputation significantly predicted high turnover intention scores (OR = 2.26, 95% CI 1.90–2.68). A higher proportion of participants who strongly agree/agree that the contributions or sacrifices made by doctors far outweigh their incomes reported significantly higher turnover intention scores in the univariable analysis but the association is not significant in the multivariable regression analysis.

## 4. Discussion

Increasing turnover intention among healthcare workers in the era of the COVID-19 pandemic has been of intense concern in China and countries badly impacted by the COVID-19 pandemic [9,14]. In China, the healthcare system was challenged by a shortage of medical doctors even before the COVID-19 pandemic. The experience of a drastic shortage of doctors during the pandemic warrants efforts to prevent turnover intention among medical students joining the health workforce upon completion of their studies.

In the descriptive analyses, the median score for fear of COVID-19 near the midpoint implies that, overall, students in this study have a moderate level of perceived fear in relation to being in a medical career during the pandemic of COVID-19. Many perceived heightened fear regarding the risk of infecting family members and fear of losing life, which was similarly reported in a recent study among medical students in Poland [32], as well as healthcare workers in China [33,34] and Poland [35]. In China, as of 26 March 2020, out of 50,006 confirmed COVID-19 cases in Wuhan, nearly 5% (2457) were healthcare workers, of whom 17 have died [36]. A study has also reported that the COVID-19 pandemic has instilled tremendous fear among frontline nurses in China, and nursing students who will soon become nurses have similarly expressed heightened anxiety [37]. In this study, slightly over half the recorded MSPSS scores of 5.01–7.00 regarding support from family, friends and significant other indicated over half of the students had high social support. The level of social support of medical students in this study is relatively higher than that of medical students in Iran [38] but lower than in Malaysia [39]. Another positive finding is that a high proportion (79.6%) reported resilience scores of 3.00–5.00 implying a great proportion having normal to high resilience. The mean BRS score in this study is near similar to those reported among medical students in U.S. [40] and undergraduate students from a university in Guangzhou, China [41].

The finding of perceived turnover intention among medical students in this study warrants serious attention. Based on the median score of 7 of the turnover intention score range of 3–15, and with approximately 58% in the high turnover score range of 7–15, the findings imply that the medical students who participated in this study scored an average turnover intention. Our finding indicates that the country may continue facing a shortage of the medical doctor workforce if no concerted effort is made by the healthcare industry and government to curb the current issue. Considering the crucial shortage of the medical workforce and particularly in the era of the COVID-19 pandemic [6], the potential of losing approximately half of medical students going into the healthcare workforce could increase the scale of distress among existing healthcare professionals resulting in a catastrophic impact on the country’s quality of health service provision. As pandemic generates enormous demand for medical staff, the country may also be likely to confront healthcare professional shortage for current or future pandemics. The multivariable regression model identified a higher year of study, perceived social status and reputation of a medical doctor, low social support and resilience, and high level of fear of COVID-19 as predictors of turnover intention, in declining predictive strength. The most important highlight of the study is the high odds of turnover intention among the final year medical students who would soon be entering the medical workforce. Recruitment and retention strategies are urgently required to focus on the final year students. Career motivational talks and role models were found to be useful to stimulate students’ intentions toward venturing into a healthcare career after completion of their studies [42], and should therefore be provided more frequently. In this study, the socio-economic variables did not significantly predict turnover intention in the multivariable analysis, however, turnover intention was significantly higher among participants from rural and lower household monthly income in the univariable analysis. Remuneration is negatively associated with turnover intention in most industries, likewise in healthcare services [14]. Students from lower socio-economic backgrounds exhibited higher turnover intention perhaps due to a desire for a higher paid profession.

Of the three attitudinal items sought, the medical students’ perceptions of the medical profession, that the perception that being a medical doctor is not a profession of high social status and reputation, significantly influence turnover intention. A considerable proportion of medical students in this study viewed being a doctor as not a profession of high social status and reputation, which likewise has been previously reported in China [14]. The sacrifices healthcare workers make, putting their lives at risk, and the high statistic of healthcare workers losing their lives battling the coronavirus should be highlighted to the public to cultivate a deeper sense of respect toward healthcare workers. Having a sense of high social status and respect, being praised for their sacrifices are important elements to enhance the sense of occupational prestige among the people in the medical profession. A high proportion viewed that the contributions made by people in the medical profession far outweigh their incomes. The government should consider revising the enumeration of healthcare workers so that their enumeration is on a par with their enormous sacrifices, workload and contribution. The long working hours and heavy workload among healthcare workers in China are well known and likewise agreed upon by the study participants. The students’ perception is in line with the finding from a recent systematic review that noted that the prevalence of burnout symptoms among doctors in China ranged from 66.5% to 87.8%, which is among the highest in the world compared to other countries [28]. Hence, there is a need for improvement in the implementation of policies to protect the well-being of people in the medical profession in China.

The multivariable findings also highlight the importance of social support in the turnover intention, a finding that is consistent with many studies on turnover intention among healthcare workers [15,16]. There is increasing awareness of the contribution of perceived social support on turnover intention. Social support serves as the major source in decreasing negative psychological reactions, such as hopelessness and depression and helps to decrease negative events in life and buffers against stress [43,44]. In this study, all three dimensions of support from family, friends and significant other have equal importance in influencing turnover intention. Hence, family and friends of healthcare workers should be made aware of their important role in providing support. A social support network among people in the medical profession to help individuals cope with stress would be useful.

The multivariable findings also revealed the importance of resilience in perceived turnover intention among medical students. Resilience has been reported to support an individual when faced with distress, allowing them to successfully bounce back [45]. The negative association between resilience and turnover intention found in this study was in concordance with many previous studies [18,19]. It has been suggested that the implementation of measures to foster resilience among healthcare workers and the provision of guidance towards positive coping at the organizational level during the COVID-19 pandemic are essential [46]. As resilience has been used as a positive coping strategy, the result of this emphasized the need for building resiliency during the undergraduate year, particularly among the final year medical students who will soon be entering the medical workforce, and those who also express tremendously high turnover intention. Currently, resilience building has not been part of the curriculum in major medical schools in China. Social resources and support were associated with resiliency against negative consequences of medical students’ mental health [47], this perhaps explains the same predictive strength of social support and resilience in turnover intention in this study.

Our findings are consistent with prior work showing that fear of COVID-19 was associated with turnover intention. Fear of COVID-19 was found to increase psychological stress levels, mental well-being, decrease job satisfaction, and hence result in higher turnover intention among frontline nurses [10]. It was suggested that organizational measures are vital to support the mental health of healthcare workers and address their fear of COVID-19 [10]. Peer and social support, psychological and mental support services, such as counselling or psychotherapy were deemed to be essential [10]. Hence, our study sheds light on the importance of addressing the fear of COVID-19 among healthcare workers.

The results of this study have important implications for practice. In the array of strength, the multivariable model shows that the strongest predictor of perceived turnover intention was being in the higher grade of study, followed by the perception that being a medical doctor is not a profession of high social status and reputation. Both resilience and the three dimensions of social support showed a similar level of predictive strength. Fear of COVID-19 was the least strong predictor of turnover intention. An important implication of this study is that the first and foremost importance is to carry out targeted intervention strategies to inhibit turnover intention among the medical students who are near completion of their studies in order to promote joining the healthcare workforce. Secondly, COVID-19 undoubtedly inflicts fear that may influence turnover intention among medical students and that the perception that being a doctor is not a profession of high social status and reputation strongly exacerbates turnover intention. Nonetheless, social support and resilience have a stronger effect than fear of COVID-19 in predicting turnover intention. Hence, our results suggest that enhancing social support at the family, community and institution level, coupled with resilience building among medical students, is recommended.

Some limitations should be noted when drawing inferences from the findings of this study. First, this study used a cross-sectional study design, which precluded the evaluation of the causality of the observed relationships. Second, data were collected from participants’ self-reports; thus, these may be subjected to socially desirable responses. Third, we queried the participants regarding their perceived level of fear in relation to being in a medical career during the pandemic approximately a year after the COVID-19 outbreak began in Wuhan in December 2019. The country has managed to control the pandemic rapidly and effectively, hence responses to fear may be subjected to response bias. Fourth, it should be noted that the turnover intention does not necessarily result in actual intention; therefore, results should be interpreted with caution. It is also important to note that only 2922 out of the total sample of 5249 medical students responded to the survey. Of these, a relatively large proportion of responses were year 1 and 2 students. Therefore, the findings of overall participant’s level of perceived fear in relation to being in a medical career during the COVID-19 pandemic, perceived social support, resilience, attitudes towards the medical profession and turnover intentions reported in this study should be interpreted with caution. Future studies should ensure inclusion of a more representative samples of students from all academic years. Last is the limitation in the sampling method, which involved recruiting participants from only one medical school in the Fujian province, thus the results of the study may not be generalizable to the medical student population as a whole.

## 5. Conclusions

The demand for healthcare workers, particularly doctors, continues to increase with the world facing the great uncertainty of the COVID-19 pandemic, and medical students are the important future healthcare workforce. Our results suggest that the COVID-19 pandemic may worsen the existing turnover of the medical workforce in China. The long-existing less privileged position of the medical doctor in society strongly stimulates turnover intention. Given that the findings imply that social support and resilience served as coping variables, enhancing resilience in medical trainees and imparting resilience building in the medical curriculum is essential. The study outlines the need for reform of current health workforce policies toward enhancing the welfare of people in the medical profession.

## Figures and Tables

**Figure 1 ijerph-18-05071-f001:**
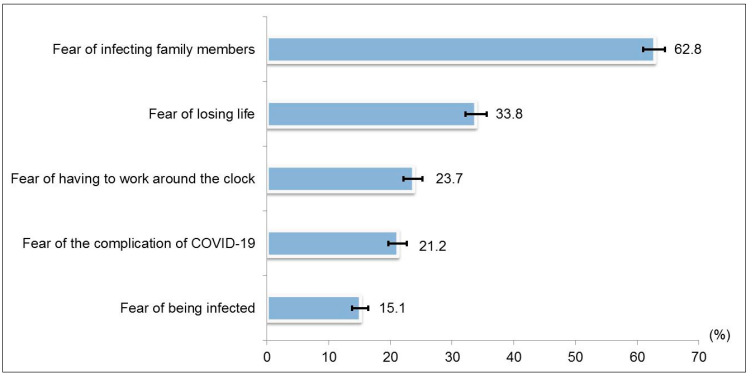
Proportion of ‘very afraid/moderately afraid’ responses for perceived fear items (N = 2922).

**Figure 2 ijerph-18-05071-f002:**
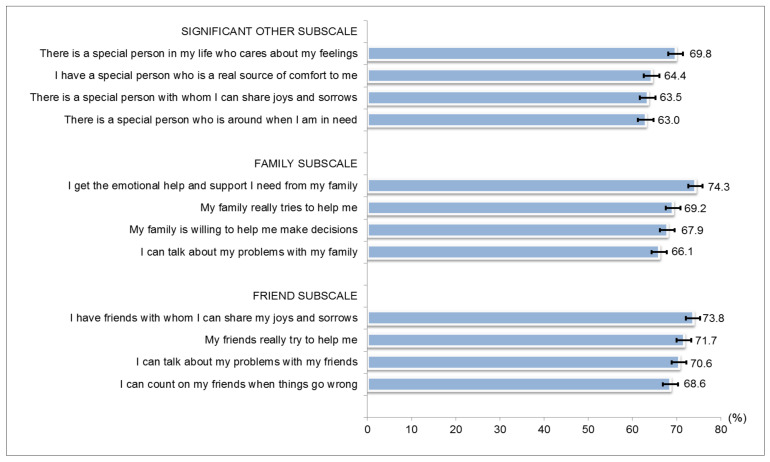
Proportion of ‘agree’ responses for multidimensional scale of perceived social support (N = 2922).

**Figure 3 ijerph-18-05071-f003:**
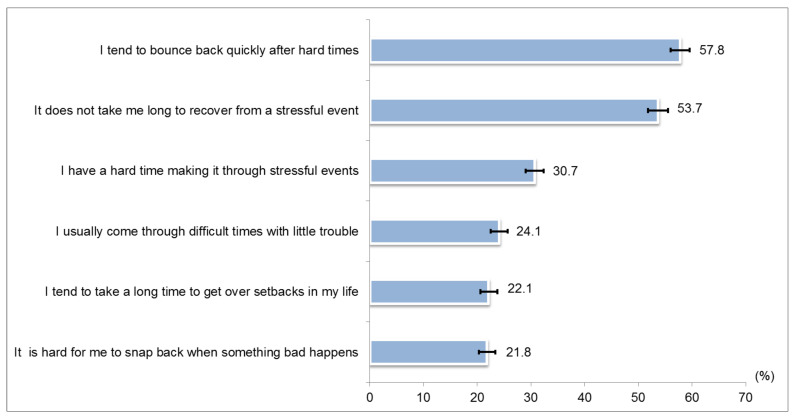
Proportion of ‘strongly agree/agree’ responses for Brief Resilience Scale items (N = 2922).

**Figure 4 ijerph-18-05071-f004:**
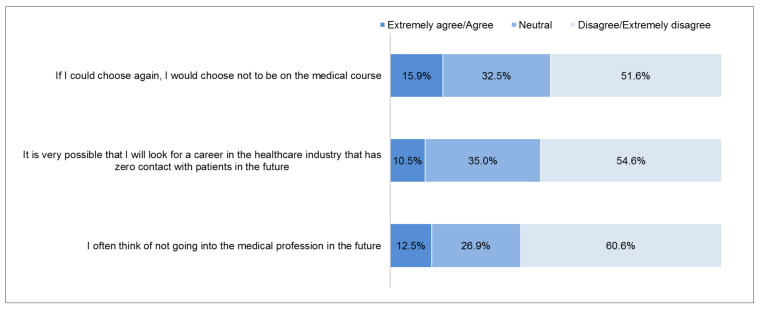
Proportion of responses for turnover items (N = 2922).

**Figure 5 ijerph-18-05071-f005:**
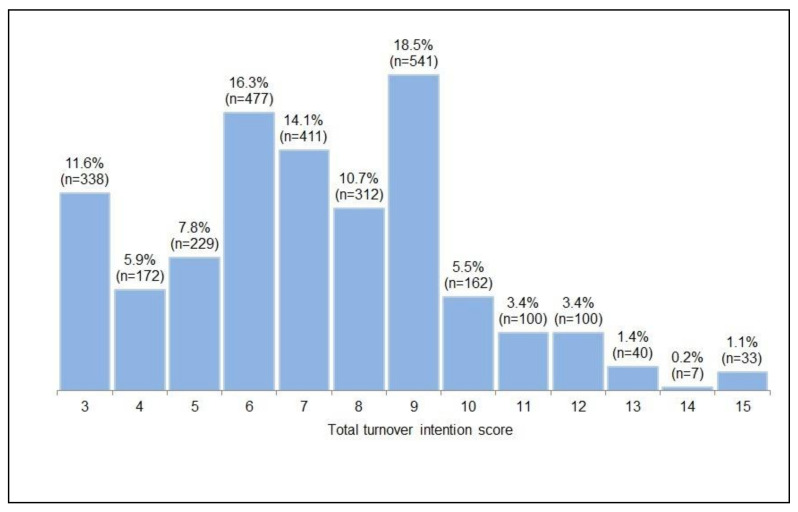
Distribution of turnover intention score (N = 2922).

**Table 1 ijerph-18-05071-t001:** Demographics of participants and factors associated with total turnover score (N = 2922).

	Overall	Univariable Analysis			Multivariable Analysis
		Total Turnover Score			Total Turnover Score 7–15 vs. 3–6
	N (%)	3–6 (n = 1216)	7–15 (n = 1706)	*p*-Value	OR (95% CI)
Gender					
Male	1367 (46.8)	547 (40.0)	820 (60.0)	0.106	
Female	1555 (53.2)	669 (43.0)	886 (57.0)		
Ethnicity					
Han	2837 (97.1)	1184 (41.7)	1653 (58.3)	0.503	
Others	85 (2.9)	32 (37.6)	53 (62.4)		
Birthplace					
Urban	1269 (43.4)	567 (44.7)	702 (55.3)	0.004	Reference
Rural	1653 (56.6)	649 (39.3)	1004 (60.7)		1.16 (0.98–1.37)
Year of study					
Year 1	793 (27.1)	449 (56.6)	344 (43.4)		Reference
Year 2	1017 (34.8)	404 (39.7)	613 (60.3)		1.72 (1.41–2.10) ***
Year 3	454 (15.5)	178 (39.2)	276 (60.8)	*p* < 0.001	1.77 (1.37–2.28) ***
Year 4	452 (15.5)	143 (31.6)	309 (68.4)		2.34 (1.80–3.03) ***
Year 5	206 (7.0)	42 (20.4)	164 (79.6)		3.88 (2.62–5.73) ***
Household monthly income (CNY)					
Less than 4000	673 (23.0)	242 (36.0)	431 (64.0)		1.07 (0.85–1.36)
4000–9999	1437 (49.2)	623 (43.4)	814 (56.6)	0.003	0.94 (0.77–1.14)
10,000 and above	812 (27.8)	351 (43.2)	461 (56.8)		Reference
Fear of COVID-19					
Total fear score					
0–5	1326 (45.4)	626 (47.2)	700 (52.8)	*p* < 0.001	Reference
6–15	1596 (54.6)	590 (37.0)	1006 (63.0)		1.39 (1.78–1.63) ***
Multidimensional Scale of Perceived Social Support (MSPSS)					
Significant other subscale					
1.00–5.00	1433 (49.0)	431 (30.1)	1002 (69.9)	*p* < 0.001	1.47 (1.17–1.84) **
5.01–7.00	1489 (51.0)	785 (52.7)	704 (47.3)		Reference
Family subscale					
1.00–5.00	1243 (42.5)	365 (29.4)	878 (70.6)	*p* < 0.001	1.47 (1.18–1.83) **
5.01–7.00	1679 (57.5)	851 (50.7)	828 (49.3)		Reference
Friend subscale					
1.00–5.00	1367 (46.8)	416 (30.4)	951 (69.6)	*p* < 0.001	1.42 (1.14–1.77) **
5.01–7.00	1555 (53.2)	800 (51.4)	755 (48.6)		Reference
Brief Resilience Scale (BRS)					
Total BRS score					
Low resilience score (1.00–2.99)	595 (20.4)	183 (30.8)	412 (69.2)	*p* < 0.001	1.44 (1.17–1.77) ***
Normal/high resilience (3.00–5.00)	2327 (79.6)	1033 (44.4)	1294 (55.6)		Reference
Attitudes towards medical profession					
Doctor is not a profession of high social status and reputation					
Strongly agree/agree	1137 (38.9)	309 (27.2)	828 (72.8)	*p* < 0.001	2.26 (1.90–2.68) ***
Strongly disagree/disagree	1785 (61.1)	907 (50.8)	878 (49.2)		Reference
The contributions and sacrifices made by doctors far outweigh their incomes					
Strongly agree/agree	2460 (84.2)	951 (38.7)	1509 (61.3)	*p* < 0.001	1.66 (1.29–2.13)
Strongly disagree/disagree	462 (15.8)	265 (57.4)	197 (42.6)		Reference
Doctors often experience long working hours with tremendous workload					
Strongly agree/agree	2704 (92.5)	1113 (41.2)	1591 (58.8)	0.086	0.83 (0.59–1.17)
Strongly disagree/disagree	218 (7.5)	103 (47.2)	115 (52.8)		Reference

** *p* < 0.01, *** *p* < 0.001. Hosmer–Lemeshow test, chi-square: 11.694, *p*-value: 0.165; Nagelkerke R^2^: 0.203.

## Data Availability

The datasets used and/or analyzed during the current study are available from the corresponding author on reasonable request.

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
