# Peer review of "Effect of the COVID-19 Pandemic on Medical Student Career Perceptions: Perspectives from Medical Students in China"

_ijerph, 2021, doi:10.3390/ijerph18105071_

Round 1

Reviewer 1 Report

The Aims of the study was to examine the perceived occupational turnover intention among medical students and 12 the associated factors. The study  results suggest that the COVID-19 pandemic may worsen the existing turnover of the medical workforce in China. The long-existing less privileged position of the medical doctor in society strongly stimulates turnover intention. The authors findings imply that social support and resilience served as coping variables, enhancing resilience in medical trainees and imparting resilience build-in the medical curriculum is essential.

 This article is important and interesting. However several methodological issues should be addressed:

  1. Low attendance to the study 2922 out of a total sample of 5249 should be added and explained in the limitation section.
  2. Was the Attitudes towards the Medical Profession scales validated before, this point should be addressed.
  3. The authors should explain and justify modification made to the version of the turnover intention assessment scale by Camman, Fichmen, 134 Jenkins and Klesh (1979), including validation of this modification.
  4. Majority of students were from years 1-2, this point should be addressed and explained in discussion and interpretation of results.
  5. Dichotomy use of subscales should be explained (E.g why division to 1-5 and 5-7).
  6. Responses to the MSPPS, BRS should be compared to other studies in similar samples.

Author Response

Please find our reply in the attached response letter.

Reviewer 2 Report

I have read your paper with great interest, however, I have few comments: 

  1. Please provide a Responsive Rate in percentages in line 158.
  2. Please provide a figure/table which will precisely determine how many students had a certain Total Turnover Score in order to make the statistical analysis reliable. 
  3. Line 252- Missing comparison about the COVID-19 related fear between the study group and the medical students and healthcare workers from other countries. There are few recently released articles in that field: 10.3390/vaccines9030218 ; 10.3390/vaccines9020128
    For example, in the group of medical students in Poland, the fear of infecting the family members was also the most significant.   
  4. In line 267 add a description of the scale of healthcare workers shortage problem, for example, https://www.ncbi.nlm.nih.gov/pmc/articles/PMC7885982/ 

Author Response

(The authors gave the same response as above.)

Reviewer 3 Report

This is a very interest paper that shows the effects of COVID-19 on medical students in China.

I have the following comments that need to be addressed in the manuscript:

  1. There is no mention of IRB or informed consent in the design or anywhere in the paper. Please clarify these two points of ethical conflict.
  2. Was the fear of COVID-19 self-developed survey questions weighted in some way that the composite score was 0-15 for 5 items with a 0-4 score? I think this would come out to 0-20 and so were some of these weighted? This needs to be clarified as this are self-developed survey questions.
  3. In the measures section, Attitudes toward the Medical Profession were 3 items of self-developed statements, 1-4 for scoring and there was not any other information. Please clarify for the readers, so that there is better veracity in the presentation of these items and the items mentioned in 2., above Fear of COVID-19.
  4. In the discussion section, new literature is introduced that should have been part of the introduction and then referred to in the discussion and supported or refuted by the results of this study. This occurs throughout the discussion section.
  5. Last in the limitation section, there is a statement about the sampling method, stating that the participants were recruited from one medical school and then stating that the “results of the study may not be generalizable to the nursing student population as a whole.” This statement needs to be changed to refer to medical student population as a whole as the study was about medical students and not nursing students. It is wholly possible that the results in a study of this type would yield completed different results when conducted on nursing students.

Author Response

(The authors gave the same response as above.)

Reviewer 4 Report

The article is interesting.
However, the research was carried out at a very early stage of the pandemic, in November and December 2020, where the ignorance and fear associated with COVID19, may have influenced the opinion of medical students.
There are other instruments that could be used.
This is a very specific reality in China, because in most countries, doctors have a high social status

Author Response

(The authors gave the same response as above.)

Round 2

Reviewer 2 Report

Dear Authors, 

I would like to congratulate you. The topic is interesting. Moreover, it is really well presented. 

Best wishes 

Reviewer 4 Report

Some relevant changes have been made.
In the introduction, the context must be characterized. Data collection took place between November 23 and December 6, 2020, so this should be contextualized as in the beginning of the pandemic, where uncertainty and fear were more significant